# Co-Stimulatory Receptor Signaling in CAR-T Cells

**DOI:** 10.3390/biom12091303

**Published:** 2022-09-15

**Authors:** Mackenzie M. Honikel, Scott H. Olejniczak

**Affiliations:** Immunology Department, Roswell Park Comprehensive Cancer Center, Buffalo, NY 14263, USA

**Keywords:** chimeric antigen receptor, T cell engineering, co-stimulation, CD28, 4-1BB, signaling, hematologic malignancies

## Abstract

T cell engineering strategies have emerged as successful immunotherapeutic approaches for the treatment of human cancer. Chimeric Antigen Receptor T (CAR-T) cell therapy represents a prominent synthetic biology approach to re-direct the specificity of a patient’s autologous T cells toward a desired tumor antigen. CAR-T therapy is currently FDA approved for the treatment of hematological malignancies, including subsets of B cell lymphoma, acute lymphoblastic leukemia (ALL) and multiple myeloma. Mechanistically, CAR-mediated recognition of a tumor antigen results in propagation of T cell activation signals, including a co-stimulatory signal, resulting in CAR-T cell activation, proliferation, evasion of apoptosis, and acquisition of effector functions. The importance of including a co-stimulatory domain in CARs was recognized following limited success of early iteration CAR-T cell designs lacking co-stimulation. Today, all CAR-T cells in clinical use contain either a CD28 or 4-1BB co-stimulatory domain. Preclinical investigations are exploring utility of including additional co-stimulatory molecules such as ICOS, OX40 and CD27 or various combinations of multiple co-stimulatory domains. Clinical and preclinical evidence implicates the co-stimulatory signal in several aspects of CAR-T cell therapy including response kinetics, persistence and durability, and toxicity profiles each of which impact the safety and anti-tumor efficacy of this immunotherapy. Herein we provide an overview of CAR-T cell co-stimulation by the prototypical receptors and discuss current and emerging strategies to modulate co-stimulatory signals to enhance CAR-T cell function.

## 1. Introduction

Pioneering work of Zelig Esshar and colleagues in the early 1990s led to the generation of the first chimeric T cell receptor (TCR), which combined the specificity of an anti-tumor antibody to TCR signaling, thus permitting T cell activation in response to any cell surface antigen of interest [1]. This was accomplished by fusing the antigen binding domain of an antibody in the form of a single chain variable fragment (scFv) to the signaling moieties of the TCR, specifically the CD3ζ chain. Such a groundbreaking advance in genetic engineering was critical to overcoming challenges faced by cellular immunotherapy for cancer, including weakly immunogenic tumors and cancer cell evasion of immune recognition [2]. Subsequent iterations and improvements in chimeric receptor design stemmed from the work of Michael Sadelain, Carl June, Renier Brentjens, and others, and ultimately resulted in the coining of the term chimeric antigen receptor (CAR) in the early 2000s and FDA approval of CD19-directed CAR-T cells for B cell ALL in 2017. Since then, CAR-T technology has dramatically expanded to encompass new targets and disease indications. Six additional FDA approvals have resulted in the span of 5 years, all of which have revolutionized treatment approaches for hematologic malignancies including various subsets of B cell lymphomas and multiple myeloma. Despite their rapid and unparalleled success, current CAR-T cell therapies have important limitations including life-threatening toxicities, finite CAR-T cell persistence and limited efficacy in solid tumors. An array of approaches in the scientific literature aim to improve outcomes of CAR-T cell therapy, including combination immunotherapy and novel genetic engineering strategies to improve the CAR design. These efforts have led to an appreciation of the importance of co-stimulatory signaling to CAR-T cell therapy [3]. This review will provide an overview of the co-receptor signals currently incorporated into CAR design and highlight the implications of these designs on several aspects of therapeutic outcomes.

## 2. Structure and Design of the Chimeric Antigen Receptor (CAR)

CARs represent a highly effective strategy to redirect T cell specificity toward a molecule of interest, most commonly an antigen that is expressed on cancer cells. The chimeric receptor is composed of several integral parts that bridge antigen recognition to T cell signaling driving T cell activation following ligation of the CAR (Figure 1). 

### 2.1. Extracellular Domains

#### 2.1.1. Antigen Targeting Domain

Conventional T cell activation requires TCR recognition of peptide residues presented within the binding groove of major histocompatibility complex (MHC) molecules on the surface of antigen presenting cells or tumor cells. CAR-T cells are equipped with synthetic machinery enabling them to recognize antigen in a non-MHC restricted manner. Antigen recognition domains of CARs are commonly derived from anti-tumor monoclonal antibodies as a result of the early findings of Zelig Eshhar that revealed similar structure and organization of the TCR and antibody molecules [4]. The specificity of the CAR is defined by the single chain variable fragment (scFv), a fusion of the variable heavy (V_H_) and light (V_L_) chains of an antibody [4,5]. ScFv molecules targeting CD19 are commonly murine derived; however, in the case of multiple myeloma, scFv molecules developed to target B cell maturation antigen (BCMA) are also of human or alpaca origin [6,7,8,9]. ScFv molecules have been designed to target a variety of cell surface molecules including proteins, carbohydrates and glycolipids overexpressed by several malignancies with much higher affinity (100×) than conventional TCRs. Several tumor antigens are currently clinical targets of CAR-T cells including the two FDA-approved target antigens CD19, and CD274 (BCMA) as well as other hematopoietic lineage markers including CD20, CD22, CD33, CD38 and CD5. CAR-T cell targets in solid tumor malignancies include Mesothelin, Prostate stem cell antigen (PSCA), Human epidermal growth factor receptor 2 (HER2), Mucin1 (MUC1), B7 homolog 3 (B7-H3), Epidermal growth factor receptor (EGFR), Ganglioside G2 (GD2) and Fibroblast activation protein-α (FAP) among others [10,11,12,13]. The V_H_ and V_L_ chains composing the scFv are joined by a flexible peptide linker sequence - most commonly (Gly_4_Ser)_3_ - that are typically 9-20 amino acids in length to allow appropriate folding and pairing of the heavy and light chains [14]. The orientation of the V_H_–linker–V_L_ can affect binding affinity and specificity of the receptor [15]. Additionally, recent data indicate a role for the linker domain in facilitating CAR surface clustering and enhanced downstream PI3K and MAPK signaling [16].

#### 2.1.2. Hinge Domain

The hinge region of the CAR connects the scFv to the transmembrane domain and is generally thought to impart a degree of flexibility to the scFv. Hinge domains are commonly composed of either Fc portions of IgG1 antibodies, Ig-like regions of the T cell co-receptors CD4 or CD8 or the T cell co-stimulatory molecule, CD28 [17,18]. The hinge helps facilitate CAR–target interactions, as a longer hinge domain may be required to access epitopes residing closer to the target cell membrane [19,20]. In addition to concerns regarding spatial flexibility and steric hindrance, the choice of hinge domain has also been implicated in various aspects of CAR-T cell functionality. Co-stimulatory receptor-derived hinge domains may promote CAR dimerization or aggregation with T cell signaling receptors as a result of the prevalence of cysteine and proline residues. It has also been shown that the hinge domain can reduce the threshold of CAR-T cell activation as evidenced by increased phosphorylation of immunoreceptor tyrosine-based activation motifs (ITAMs) within the CD3ζ chain and increased production of effector cytokines including interferon-γ (IFN-γ) and tumor necrosis factor-α (TNF-α) [17,18,21]. 

Current evidence demonstrates that small changes to the extracellular structure of the CAR can have large consequences on signaling and ultimately therapeutic efficacy. Given the hundreds of CARs currently in clinical and preclinical use, none of which are identical, identifying the effect of any one domain on CAR-T cell function is confounded by differences in other domains in the construct. However, this highlights the need to precisely and strategically study CAR engineering to fully understand this synthetic biological system. 

### 2.2. Intracellular Domains

#### 2.2.1. Transmembrane Domain

The primary function of the transmembrane domain is to anchor the CAR to the T cell membrane and propagate ligand recognition signals to the cytoplasmic signaling portion of the receptor. Transmembrane domains are commonly derived from membrane spanning portions of T cell co-receptors CD4 or CD8, or the co-stimulatory molecule CD28 [22]. In recent years, the field has begun to unveil how the origin of the transmembrane domain contributes to CAR signaling. A core of hydrophobic residues within the CD28 transmembrane region have been shown to facilitate physical interactions between the CAR and endogenous CD28 co-stimulatory receptor to form heterodimers [23]. However, these interactions are disrupted when a CD8α transmembrane domain is incorporated. Recent work has shown that the inclusion of a CD28 transmembrane reduces the threshold of antigen density required for T cell activation [21], causes antigen-independent activation and proliferation as a result of constitutively active signaling [24], and yields elevated levels of IFN-γ secretion [17]. Collectively these reports reinforce the notion that CAR and TCR-associated signaling molecules can interact to further accelerate and prolong the duration of CAR signaling. Such signaling may be detrimental to in vivo CAR-T efficacy and promote toxicities associated with overactive CAR-T cells. The ability of a CAR to associate with endogenous receptors adds complexity to the conceptual framework surrounding CAR-T cells signal transduction, and also highlights additional elements that should be considered when optimizing CAR design.

#### 2.2.2. T Cell Activation Domain

Signaling components of the CAR-enabling T cell activation compose the cytoplasmic fraction of the receptor. Analogous to the “two-signal” model of T cell activation, the membrane distal CD3ζ chain contributes “signal 1” of T cell activation, while the membrane proximal co-stimulatory domain provides “signal 2”. CD3 is the signaling component of the TCR complex. The CD3ζ chain contains three immunoreceptor tyrosine-based activation motifs (ITAMs) within its cytoplasmic tail consisting of highly conserved Tyr-X-X-Leu X_6-8_ Tyr-X-X-Leu sequences [25]. Interactions between the TCR and MHC-peptide complex —stabilized by T cell co-receptors CD4 or CD8 —lead to the activation of Lck, a Src family tyrosine kinase, which is held inactive by co-receptors CD4 and CD8 prior to TCR engagement. Lck phosphorylates tyrosine residues present within the ITAMs, leading to the recruitment and subsequent activation of ZAP-70. ZAP-70 in turn phosphorylates a number of substrates including LAT, SLP-76 and PLC-γ. These events catalyze the activation of downstream signaling pathways required for changes in gene expression to facilitate effector T cell responses [25,26,27]. Lck also associates with and is activated by co-stimulatory CAR domains leading to subsequent phosphorylation of ITAMs present in the CD3ζ domain. However, the intricate details of Lck activation to induce CAR signaling is still an active area of investigation [28,29,30].

The CD3ζ chain has been the focus of recent research efforts aimed at dampening activation signaling in CAR-T cells. Prior work demonstrated that continuous T cell activation signaling, or chronic antigen stimulation drives terminal T cell differentiation and exhaustion leading to a CAR-T cell product with limited persistence, proliferative potential, and anti-tumor efficacy [31,32,33]. Work by Michel Sadelain’s group demonstrated that redundancy in signaling initiated by ITAMs within the CD3ζ chain of the CAR drove counterproductive T cell differentiation and exhaustion. They subsequently mutated individual ITAMs at varying positions to find that a single functional ITAM positioned at the membrane proximal location was sufficient for a potent anti-tumor response and facilitated differentiation into long-lived central memory-like T cells with preservation of the expression of memory associated transcription factors including transcription factor 7 (TCF7), B-cell lymphoma 6 protein (BCL6) and lymphoid enhancer-binding factor 1 (LEF1). ITAM-mutated CAR-T cells displayed improved persistence in a murine leukemia model and response to tumor re-challenge, indicating preservation of effector T cell function [34]. Altogether these findings support a model of CAR-T cell fate that is dependent upon the strength of signaling received and the positioning of signaling elements within the receptor. 

#### 2.2.3. Co-Stimulatory Domain

First generation CAR-T cells lacked a co-stimulatory domain yet demonstrated cytolytic capacity in preclinical studies, including eliciting anti-tumor responses in murine tumor models [35,36,37,38]. However, their success in patients was limited, likely due to dramatic contraction of the infused CAR-T population in the days following infusion [39,40]. Inclusion of a co-stimulatory signal in CAR-T cells began with the work of Renier Brentjens and Michel Sadelain in the early 2000s [3,41,42,43,44] based on recognition that co-stimulation is required for T cell expansion and IL-2 production [45,46] along with evasion of T cell anergy [47] and apoptosis [48]. Modifications of first generation CARs to include CD28 or 4-1BB co-stimulatory domains led to the development of second generation CAR-T cells, which currently comprise all FDA-approved CAR-T cell products [49,50,51,52,53,54,55,56,57]. Inclusion of a co-stimulatory domain drastically improved CAR-T cell performance in clinical trials and enhanced CAR-T persistence in the body following infusion. Impressively, a recent report highlighting the first patients treated with second generation CAR-T cells nearly a decade prior remain in remission and illustrate a sustained CD4^+^ CAR-T cell population with a cytotoxic and functionally active phenotype [58].

Due to the success of CD28 and 4-1BB containing second generation CAR-T cells, current research efforts are exploring attributes of additional co-stimulatory domains—including ICOS (CD278), OX40 (CD134) and CD27—or the combination of multiple co-stimulatory domains, as in the case of third generation CAR-T cells [59,60,61,62,63]. Third generation CAR-T cell products incorporate two co-stimulatory domains within the cytoplasmic signaling tail with the attempt of providing added benefits and overcoming limitations of any one co-stimulatory signal [64]. Third generation, CD19-targeted CAR-T cells have been reported to show greater expansion, survival and persistence in both preclinical and clinical settings; however, an enhancement in anti-tumor activity and clinical efficacy of third generation CAR-T cells over second generation designs remains controversial in the field [60,61,63,65,66,67,68,69,70] Furthermore, the tumor type, preclinical model and preconditioning regimen employed when evaluating varying CAR-T cell generations adds additional layers of complexity. However, there remain active and ongoing investigations into third generation CAR-T cell products to better understand the consequences of additive co-stimulatory signaling in engineered T cells. 

The choice of co-stimulatory domain included in the CAR can greatly impact several key aspects of CAR-T cell therapy such as the differentiation state of the infusion product, the kinetics of anti-tumor response, cytotoxic function and associated toxicities. Recent work has proposed that an optimized co-stimulatory signal can generate a more desirable CAR-T cell product with an extended in vivo life span [59]. This notion stems from prior evidence showing that excessive co-stimulation is detrimental to CAR-T cell performance [71,72], and that mutation of specific residues within the cytoplasmic tail of a CAR that disrupt downstream signaling enhanced anti-tumor efficacy and CAR-T persistence in vivo [60,73,74]. Further complicating matters, CAR-T cells retain surface expression of endogenous TCRs and co-stimulatory receptors that can be activated by cognate ligands in the tumor microenvironment. Several studies have shown that CARs and endogenous co-stimulatory receptors such as CD28 can heterodimerize, potentially leading to reinforced signal transduction and/or tonic CAR-T cell activation in the absence of target antigen [17,21,23,24,75,76]. 

Thus far, our understanding of CAR signaling remains underdeveloped, with multiple intertwined signaling pathways currently implicated. Understanding the complexities of CAR-T cell signaling will facilitate future targeted improvements to CAR design to achieve desired biological outputs that allow CAR-T cells to systemically eradicate cancer. The remainder of this review focuses on the impact of co-stimulatory receptors in CAR-T cells, specifically the alterations that have been employed and the implications of these modifications on signaling at the molecular level as well as on various aspects of CAR-T cell functionality.

## 3. Co-stimulatory Receptor Signaling Pathways

The work of multiple different groups over the course of the past several years has focused precisely on the co-stimulatory element of CAR-T cells. Unsurprisingly, the choice of co-stimulatory moiety encoded within the CAR has led to drastically different functional attributes among CAR-T cell products, due in part to the differences in the signaling mechanisms between endogenous T cell co-stimulatory receptors. Such differences have also been observed at the molecular level in various second-generation CAR-T cell designs through the evaluation of signal transduction pathways [60,61,71,76]. Here we review the signal transduction pathways activated by endogenous T cell co-stimulatory receptors that have been incorporated into CAR design, highlighting differences that may account for alterations in CAR-T cell functionality.

### 3.1. Immunoglobulin Superfamily

#### 3.1.1. CD28

CD28 was identified in the late 1980s as the first surface receptor with T cell co-stimulatory function [77]. It is the classical, most well described co-stimulatory molecule required for naïve T cell activation and is constitutively expressed as a homodimer on the surface of the majority T cells in young individuals, with a decrease in CD28 expressing T cells observed with advanced age. The ligands for CD28, B7 family molecules B7-1 (CD80) and B7-2 (CD86), are present on the surface of professional antigen presenting cells. CD80/86 ligation of CD28 contributes a positive co-stimulatory signal during T cell priming by both amplifying the magnitude of TCR driven signals and mediating a unique set of signaling events that influence T cell survival and proliferation [48,78,79], metabolism [80], epigenetic reprogramming [81,82,83], cytokine production [77,84,85,86,87,88], cytoskeletal remodeling [89,90] and RNA splicing [91,92]. Details of CD28 function during T cell activation have been reviewed previously [93].

Dissection of the CD28 signaling pathway led to the elucidation three motifs within its short cytoplasmic tail, all of which have been experimentally targeted in the chimeric co-stimulatory domains of CAR-T cells [73,74]. The membrane proximal tyrosine-rich motif (YMNM), the proximal proline-rich motif (PRRP) and the distal proline-rich motif (PYAP) each initiate downstream signaling through recruitment of specific adaptor proteins, which will be discussed herein, however, there is significant overlap and potential for interaction within downstream signaling pathways. 

The CD28 cytoplasmic tail has no intrinsic enzymatic activity, however, crosslinking of the receptor induces phosphorylation of the tyrosine residue of the YMNM motif by Src family kinase, Lck. Tyrosine phosphorylation causes the direct binding of the p85 subunit of PI3K. Activation of PI3K catalyzes the production of phospholipid products phosphatidylinositol (3,4)-bisphosphate (PIP_2_) and phosphatidylinositol(3,4,5)-triphosphate (PIP_3_) which act as second messengers to recruit cytoplasmic proteins to specific cell membrane locations. Phosphoinositide-dependent kinase 1 (PDK1) and Protein Kinase B (PKB)/Akt bind to PIP3 via the pleckstrin homology domain (PH) domain and subsequently phosphorylate several downstream targets including glycogen synthase kinase-3 β (GSK3β), mammalian target of rapamycin (mTOR), inhibitor of nuclear factor-kB (IκB) and regulators of cell survival, namely Bcl-2 antagonist of cell death (BAD). Activation of mTOR and IκB lead to transcriptional activity of NF-κB while GSK3β promotes transcription of NFAT-dependent genes [93,94,95]. The proximal tyrosine motif also serves as a binding site for adaptor molecules Grb2 and GADS which initiate the formation a multi-protein complex composed of SLP76, Vav and LAT. Formation of this complex results in the generation of a platform for the recruitment and activation of the PLCγ1 enzyme which hydrolyzes PIP_2_ into diacyl-glyercol (DAG) and IP_3_ which activate PKCθ and the phoshphatase, calcineurin, through the release of intracellular Ca^2+^, respectively. PKCθ regulates activation of NF-κB through phosphorylation of adaptor protein CARMA-1 and recruitment of the Bcl10-MALT-1 complex. Additionally, calcineurin activation by increased cytoplasmic Ca^2+^ levels mediates the de-phosphorylation and activation of NFAT permitting nuclear localization and transcription of pro-survival factors such as Bcl-x_L_ and cytokines such as IL-2 to sustain T cell proliferation [96,97,98,99]. 

The proximal proline-rich motif, PRRP, has been shown to associate with IL-2 inducible T cell kinase (Itk) following its activation by Lck. Once activated, Itk undergoes auto-phosphorylation leading to the phosphorylation of downstream targets including PLCγ1. In addition to inducing intracellular Ca^2+^ flux, PLCγ1 activation initiates the Erk signal transduction pathway resulting in transcription of genes associated with cell growth and proliferation [74,100,101]. 

Signaling by the distal proline-rich PYAP motif is thought to initiate a critical, non-redundant signaling pathway that is essential to T cell proliferation and cytokine production. Initiation of signaling requires the binding of Lck via the SH3 domain to the proline motif. Kinase activity of Lck results in the phosphorylation of PDK1 which ultimately inactivates the inhibitory GSK3β leading to the transcription of NFAT dependent genes. Grb2 also binds to the distal PYAP motif through its SH3 domain to catalyze an independent signaling cascade through formation of a complex with two guanine nucleotide exchange factors, Sos and Vav. Initiation of this complex results in cytoskeletal rearrangement through the activation of Ras, Rac1 and CDC42 and downstream MAPK activation which culminates in activation of the terminal kinase effector, JNK. JNK activation induces the formation of the AP-1 transcriptional complex composed of members of the FOS, JUN, MAF and ATF sub-families (Figure 2) [79,95,99].

Despite overlap in the adaptor molecules and crosstalk amongst signaling molecules in the CD28 pathway, various functions have been shown to be dependent on specific motifs. The YMNM motif is critical for upregulation of Bcl-x_L_ to sustain cell survival and also regulates nutrient uptake and glycolytic metabolism to meet the energetic demands of a proliferating cell [78,80,102,103]. The PYAP motif, however, has been previously shown to be critical to T cell proliferation and IL-2 production post-αCD3 and αCD28 activation and in the execution of a T cell response in the context of autoimmune disease. [99,104].

#### 3.1.2. Inducible T Cell Co-Stimulator (ICOS)

ICOS is another member of the Ig superfamily expressed at low levels on the surface of naïve T cells and upregulated upon TCR crosslinking plus CD28 stimulation [105,106,107]. Despite similar structures, ICOS binds exclusively to its ligand, ICOS-L which is expressed on professional APCs, B cells, macrophages and lung epithelial cells. Its main function is to promote the generation of T follicular helper cells, which support B cell affinity maturation within germinal centers. Additionally, ICOS contributes to anti-tumor T cell responses and graft-versus-host disease by reinforcing expression of various Th1 and Th2 cytokines such as IL-4, IL-10, IL-21 and IFN-γ.

The cytoplasmic signaling domain of ICOS resembles that of CD28, containing a similar proximal motif (YMFM) which, upon receptor ligation, recruits the regulatory (p50α and p85α) and catalytic (p110δ) subunits of PI3K [108,109]. As described in the previous section, PI3K activation results in the activation of Akt and subsequent transcription of NF-κB dependent genes which promote survival and proliferation. PI3K activation can also activate the Ras-MAPK pathway, although ICOS-mediated JNK phosphorylation is much weaker as compared to CD28-driven signaling [110,111]. In contrast to CD28, ICOS lacks the asparagine residue that recruits the adaptor molecule, Grb2. This difference in signaling has been attributed functionally to the inability of ICOS to induce IL-2 production [112]. 

Additional functions of ICOS independent of PI3K signaling have recently led to the identification of cytoplasmic motifs, further distinguishing ICOS from CD28 [113]. The ICOS proximal motif (IProx) represents one motif that recruits binding of a serine/threonine kinase, TBK1 (TANK-binding kinase 1), a non-canonical member of the IKK family. Functionally, the IProx motif is required for proper T_FH_ cell differentiation [114]. An additional motif (KKKY) has been shown to be essential to intracellular Ca^2+^ release through the activation PLCγ1 and dynamic cytoskeletal remodeling [115]. 

### 3.2. TNF-R Superfamily

Although CD28 represents the prototypical T cell co-stimulatory molecule critical to proper T cell priming, CD28 signaling is not sufficient to drive the diverse fates of activated T cells. Co-stimulation through the tumor necrosis factor receptor (TNF-R) superfamily augments T cell activation signals and plays critical roles in T cell differentiation and memory formation. TNF-R co-stimulatory molecules share substantial overlap in signal transduction pathways they stimulate and are induced by TCR stimulation, with surface expression peaking several days after antigen stimulation. As such, this inducible family of receptors function to sustain the T cell response and drive signaling critical for memory T cell development.

#### 3.2.1. 4-1BB

To date, the majority of FDA-approved CAR-T cell products incorporate a co-stimulatory signal derived from 4-1BB, a TNF-R superfamily member, as a result of the desirable attributes that this signaling moiety provides to the infused CAR-T population. 

Unlike CD28, 4-1BB (CD137, TNFRSF9) is preferentially expressed on the surface of activated T cells [116,117], and functions as an accessory molecule during the process of T cell activation [118]. Downstream signaling cascades are initiated upon ligation by its sole ligand, 4-1BB ligand (4-1BBL), which exists as a homotrimer on the surface of B cells, DCs and macrophages. Activation of the 4-1BB pathway promotes T cell proliferation through the regulation of cyclin-dependent kinases, sustains survival of activated T cells through expression of anti-apoptotic molecules and contributes to cytokine production [116,119,120]. 

Initiation of signal transduction by TNF-R receptors is mediated by a conserved class of scaffolding molecules known as TNF-R associated factors (TRAFs) [121]. 4-1BB signaling in T cells relies on the recruitment of TRAF1, 2 and 3 to assemble the signalosome and initiate signal transduction [122]. TRAF molecules bind to the intracellular tail of 4-1BB as trimers or heterotrimers at specific motifs, namely Thr-Thr-Gln-Glu-Glu and Pro-Glu-Glu-Glu-Glu within TRAF-binding consensus regions [123]. TRAF1 has no intrinsic enzymatic activity. Rather it has been implicated in the regulation of TRAF2 stability and has been identified as a critical mediator of T cell survival downstream of the 4-1BB signaling axis [60,124,125]. TRAF2 association with the 4-1BB tail is a critical event mediating the activation of downstream signaling pathways culminating in the activation of transcription factors NF-κB and AP-1 [123,126]. The RING domain-containing TRAF2 is a E3 ubiquitin ligase, which mediates the K63-linked polyubiquitination of RIP1, a serine/threonine protein kinase. K63-ubiquitinated RIP1 recruits transforming growth factor-β-activated kinase 1 (TAK1), which in turn activates IKKβ, a component of the IKK complex. The activated IKK complex subsequently phosphorylates IκB, targeting it for ubiquitin-dependent degradation by proteasomes and releasing NF-κB subunits p50 and RelA from their inhibited state. Freed NF-κB subunits translocate to the nucleus where they drive expression of pro-survival members of the Bcl-2 family of anti-apoptotic proteins, survivin, and cell cycle regulators to promote T cell survival and proliferation. In parallel, TRAF3 recruitment to 4-1BB activates the non-canonical NF-κB pathway. TRAF3 serves as an adaptor protein linking the ubiquitin ligase complex of TRAF2 and cellular inhibitors of apoptosis proteins (cIAP1 and cIAP2) to the NF-κB inducing kinase (NIK) [123]. Ubiquitination of TRAF3 induces its degradation and results in stabilization of NIK. NIK activates IKKα and induces nuclear translocation of p52 and RelB for sustained NF-κB activation [127,128]. 

In addition to its roles in canonical and non-canonical NF-κB activation downstream of 4-1BB, TRAF2 has also been shown to activate mitogen-inducible protein kinases (MAPKs), including MAP/extracellular signal-related kinase kinase 1 (MEKK1) and apoptosis signal-regulating kinase 1 (ASK-1) [129,130]. MEKK1 activates the p38 MAPK downstream signaling cascades leading to transcriptional activation of ATF2, which has been implicated in mitochondrial biogenesis [131]. ASK-1 mediates induction of the JNK/ Stress Activated Protein Kinase (SAPK) pathway culminating in activation of the AP-1 transcriptional complex. Collectively these pathways have been shown to enhance the production of cytokines and effector molecules including IL-2, IL-4, IL-5 and IFN-γ downstream of 4-1BB (Figure 2) [120,132,133,134].

#### 3.2.2. OX40

OX40 (CD134, TNFRSF4) is an additional member of the TNF-R superfamily originally identified in the late 1980s [135] with expression on CD4^+^ and CD8^+^, Th1, Th2, Th17 and FOXP3^+^ regulatory T cells. Translation of the OX40 encoding mRNA is dependent on TCR engagement and IL-2R signaling and therefore expression is restricted to activated T cells. OX40 typically appears on the T cell surface within 24–48 h post activation [126,136,137]. The ligand, OX40L shares sequence homology with other molecules within the TNF family and exists on the surface of activated APCs including B cells, DCs, macrophages, NK cells and vascular endothelial cells which are often situated at sites of inflammation. OX40L is a membrane-bound trimer with the ability to bind three OX40 monomers to initiate signaling. The most recognized functions of OX40 include its role in regulating clonal expansion through expression of pro-survival molecules and cell cycle regulators and mediating T cell differentiation and maintenance of a memory T cell pool following antigen encounter [138,139,140].

OX40 is capable of initiating signals through the recruitment of scaffolding molecules TRAF2, 3 and 5 to the cytoplasmic tail, which activates both canonical and alternative NF-κB transcriptional activity [126,141,142]. TRAF molecules initiate the formation of the OX40 signalosome which includes members of the IKK kinase complex, as well as the p85 subunit of PI3K. PI3K in turn drives PKB/Akt activation to promote cell growth, survival and cell cycle entry [143]. OX40 also has the capacity to enhance intracellular Ca^2+^ flux through the activation of the phosphatase, calcineurin, which leads to NFAT translocation and cytokine production including IL-2, although the precise mechanism by which this occurs remains incompletely understood [144]. OX40 has also been shown to drive expression of the IL-2Rα chain (CD25) which amplifies activation of PKB/Akt and initiates the JAK/STAT pathway sustaining IL-2 responsiveness in T cells [137,145]. Additional cytokine receptors including IL-7Rα (CD127) and IL-12Rβ2 have been shown to be upregulated upon OX40 ligation which may contribute to its role in T cell differentiation [145,146,147]. Overall, OX40 mediated signaling directly promotes sustained T cell survival and expansion but can indirectly regulate effector differentiation through modulation of cytokine production and cytokine receptor expression.

#### 3.2.3. CD27

Unlike most members of the TNF-R superfamily, CD27 (TNFRSF7) is expressed on the surface of resting T cells and additional lymphoid cells including B cells and NK cells, and is further induced upon activation, suggesting a more critical role for this receptor during early stages of priming. CD27 is expressed as a homodimer on the T cell surface and engages with its cognate ligand, CD70 to form higher order oligomers. Expression of CD70 is restricted to DCs, B cells and NK cells, and is vastly upregulated by pathogenic stimuli, such as Toll-like receptor activation, IL-1α, IL-12, TNFα or prostaglandin E_2_ stimulation [148,149,150]. CD27 is functionally similar to other members of the TNF-R family. TRAF molecules (TRAF2, 3, 5) directly associate with the conserved C-terminal TRAF domain within cytoplasmic tail of CD27 to mediate the activation of NF-κB and JNK to support T cell survival, proliferation and differentiation [151,152,153,154,155]. CD27 similarly results in activation of both canonical and alternative NF-κB pathways through the formation of a multi-protein complex initiated by TRAF2 which recruits cIAP family molecules, NIK, and the IKK kinase complex [151]. Upon TCR stimulation, CD27 fosters T cell survival through the expression of anti-apoptotic molecules (Bcl-xL) to sustain clonal expansion while also down-regulating expression of FasL to protect T cells from undergoing FasL-induced cell death [156,157]. CD27 largely promotes effector differentiation by protecting cells from undergoing apoptosis during expansion, but is not critical to cytotoxic functions, such as IFN-γ production, of terminally differentiated effector cells and is cleaved from the effector cell surface [153,158]. Additionally, several investigators have illustrated a critical role for CD27 in the generation of a memory T cell population with the capacity to respond to a secondary challenge, largely in the context of viral infection models [153,159,160,161,162]. CD27 co-stimulation influences several aspects critical to a proper T cell-mediated immune response and has demonstrated a prominent role in sustaining survival of a central memory T cell population. The desirable attributes discussed above have led to CD27 signaling domain incorporation into CAR design in hopes of building a longer lasting CAR-T cell product.

## 4. Functional Implications of Co-Stimulation

Second generation CAR-T cell products have elicited tremendously exciting, curative responses in patients with hematologic malignancies. Several studies have evaluated the role of the chimeric co-stimulatory signal on multiple aspects of CAR-T cell therapy, including anti-tumor CAR-T cell response, persistence, phenotype and associated toxicity profiles. Below we provide an overview of the preclinical and clinical evidence demonstrating the influence of co-stimulatory domains on various facets of CAR-T cell function. Unfortunately, findings across clinical trials and investigational studies remain confounded by several variables including differences in CAR components and expression systems, ex vivo culture conditions, and lymphodepleting regimens to name a few Despite these differences, generalizable conclusions regarding effects of co-stimulatory domains are widely accepted in the field.

### 4.1. Phenotype

Classification of various T cell subsets including naïve, memory stem, central memory, effector memory and effector, reflect different stages of T cell differentiation in response to antigen stimulation. Although the pathways that drive T cell commitment to a specific cell fate remain a topic of debate in the field of immunology, the subsets are largely defined by the phenotypic expression of cell surface molecules, gene expression profiles, chromatin structure and metabolic programs. Since the advent of adoptive T cell therapy, there has been increased focus on functional characteristics of T cell subsets, including proliferative capacity and anti-tumor efficacy, with the goal of generating an effective and durable T cell infusion product. 

#### 4.1.1. T Cell Differentiation

Seminal work in the field of adoptive immunotherapy demonstrated that less differentiated T cell subsets (naïve/stem memory/central memory) elicit greater tumor control as compared to terminally differentiated effector cells [163,164,165,166]. Less differentiated T cells can give rise to daughter cells that carry out cytotoxic functions, while also replenishing the memory population for a longer response duration. Terminally differentiated effector cells demonstrate limited in vivo survival, cytokine production and proliferative burst. The distinction between T cell subsets usually relies on differential surface expression of specific molecules including CD45RA, CD45RO, CD62L, CCR7, CD27, and IL-7Rα that dictate T cell trafficking patterns of distinct T cell subsets and their response to chemokines and cytokines [167]. 

In recent years, the field of CAR-T cell therapy has sought potential ways to skew CAR-T cell differentiation toward a central memory or stem memory infusion product to replenish the effector T cell pool [168]. A variety of strategies involving modulation of exogenous cytokines used during CAR-T expansion, epigenetic reprogramming agents, and small molecule inhibitors to arrest the T cell differentiation process have been explored [169,170]. Choice of co-stimulatory domain also influences the phenotype and functional characteristics of CAR-T cells. Pre-clinical investigations demonstrate that the incorporation of a 4-1BB co-stimulatory domain preserves a greater frequency of central memory CAR-T cells defined as (CD45RO^+^ CD45RA^−^, CD62L^+^, CCR7^+^) as compared to CD28ζ-containing CAR-T cells, which were enriched for effector memory phenotype cells [60,171,172]. Recent work has provided mechanistic insight into these differences, linking strength of signal initiated by CD28 vs. 4-1BB with differentiation status of the CAR-T product [71]. Preclinical evidence surrounding ICOS-containing CAR-T cells demonstrates that the inclusion of the ICOS domain results in a T_H_17/T_H_1 bipolarization of the CAR-T cell product which attributes an inherent multipotency, and greater self-renewal capacity as compared to CD28 and 4-1BB containing CAR-T cells. The expression of transcriptional regulatory Rorc and Il1r1 were inherently linked to sustained CAR-T cell proliferation and survival positioning ICOS containing CAR-T cells as strong candidates for long-term anti-tumor function [169,173]. 

Analogous to investigations into 4-1BBζ CAR-T cells, the inclusion of TNF-R superfamily molecules CD27 and OX40 in CAR design supports improved antigen-dependent memory formation, and enhanced T cell survival [174,175,176]. These findings have been influential in the design of several third generation CAR-T cell constructs which combine both immunoglobulin and TNF-R superfamily molecules into a single cytoplasmic domain to achieve the desirable attributes of each of the signaling molecules (Table 1) [177,178,179] 

T cell exhaustion represents a progressive state of differentiation that ultimately leaves T cells incapable of properly responding to stimulation. In the context of chronic infection and cancer where T cells receive continuous stimulation through their TCRs, T cells experience progressive loss of proliferative potential and cytokine production, inability to degranulate, and upregulation of inhibitory molecules (PD-1, CTLA4, LAG-3, TIM-3) [31]. In CAR-T cells antigen-independent tonic signaling through the CAR drives a dysfunctional/exhausted state. In the absence of scFv ligation, CARs are capable of dimerizing to initiate a tonic, basal level of signaling which induces proliferation, cytokine production and initiation of exhaustion pathways [24]. The choice of co-stimulatory domain has been implicated in the onset of tonic CAR signaling induced exhaustion, which occurs more robustly in CD28ζ than ICOSζ, BBζ, OX40ζ or CD27ζ CAR-T cells [73]. More specifically, CD28 augments the onset of early exhaustion which limits the associated anti-tumor efficacy and is in part responsible for their short-lived persistence in vivo. However, the inclusion of a 4-1BB co-stimulatory domain ameliorated the onset of CAR-T cell exhaustion driven by tonic, antigen-independent scFv clustering and downstream signal activation. 4-1BB contributed an anti-exhaustive effect which was evident at both the transcriptional and phenotypic level through diminished expression of exhaustion related genes and negative regulation of apoptosis [32]. 

Although the precise mechanism linking CD28 to exhaustion is not inherently clear, Ca^2+^ mediated NFAT activation is thought to be involved in the regulation of anergy- and exhaustion-associated genes [73]. Elegant mechanistic studies have demonstrated that exhausted CAR-T cells have widespread epigenomic dysregulation of AP-1 transcription factor binding motifs. IRF and BATF family transcription factors are capable of forming immunoregulatory transcriptional complexes which drive expression of exhausted-associated genes. Overexpression of c-Jun prevented exhaustion by displacing the immunoregulatory complexes from chromatin in a tonically signaling CD28ζ CAR-T cell model [180]. Collectively these findings support the notion that intrinsic differences in the signaling pathways among co-stimulatory receptors may play critical roles in the differential activation of exhaustion pathways.

At this time, the cause of CAR-T cell exhaustion remains incompletely understood and is further complicated by the extensive array of CARs in clinical and preclinical use. The choice of promoter in viral expression vectors, level and duration of surface expression, strength of signaling and additional structural design elements of the CAR have all been implicated in tonic signaling and effectively combating CAR-T cell exhaustion will likely require a multi-pronged approach [24,34,71,181,182]. 

#### 4.1.2. Metabolic Profile

The metabolic state of T cell subsets varies greatly to support functional demands over the course of an immune response. Recent studies have demonstrated that T cell metabolic pathways support T cell fate by regulating differentiation into memory and effector subsets and have been comprehensively reviewed elsewhere [183,184,185,186]. Briefly, naïve T cells are metabolically quiescent; however, upon activation T cells drastically increase in size and undergo a metabolic transition from reliance on oxidative phosphorylation to aerobic glycolysis as their main source of energy (ATP). This dynamic shift is accentuated by CD28 co-stimulation [80]. Activated T cells rely on glucose and glutamine to fuel glycolysis and the citric acid cycle, respectively, in order to generate ATP and biosynthetic intermediates required for cell growth and proliferation. Upon resolution of an initial T cell response, many effector cells undergo activation-induced cell death and a small portion of T cells differentiate into memory T cells characterized by a primed metabolic state that relies on fatty acid oxidation and mitochondrial metabolism to sustain energetic and survival requirements [184,187]. 

Previous work demonstrated a role for co-stimulatory receptors and downstream adaptor molecules in priming T cell metabolism. Specifically, the ability of CD28 to enhance glycolysis in activated T cells by inducing surface localization of glucose transporters and expression of glycolytic enzymes is well documented [80,92,188,189]. Additionally, CD28 signaling has been shown to support memory T cell differentiation through mitochondrial cristae remodeling and increased mitochondria respiratory capacity, enabling rapid response to secondary stimulation [190]. 4-1BB stimulation has also been shown to induce mitochondrial biogenesis through activation of p38 MAPK to support heightened respiratory capacity in preparation for future re-activation [131]. Previously identified roles of co-stimulatory molecules in metabolic programming have catalyzed investigations into understanding the influence of chimeric co-stimulatory domains on CAR-T cell phenotype and metabolic state which greatly impact long-term therapeutic capacity in an immunotherapeutic setting. 

Recent studies have unveiled striking differences in the metabolic state of CD28ζ and BBζ containing CAR-T cells. Inclusion of the 4-1BB domain drastically enhanced respiratory capacity of the CAR-T cell population and supported mitochondrial biogenesis, as evidenced by greater mitochondrial mass relative to CD28ζ CAR-T cells. Analogous to the role of the endogenous 4-1BB receptor, the chimeric 4-1BB domain programed CAR-T cells to rely heavily on oxidative metabolism. Contrastingly, CD28ζ CAR-T cells demonstrate an increased reliance on aerobic glycolysis to support energetic needs [172]. This work also supports the notion of an interconnected relationship between the metabolic landscape and the fate of CAR-T cells, as CD28ζ cells were predominantly of effector-memory phenotype while 4-1BBζ cells display a central memory phenotype [172]. Investigations into the metabolic landscape of CAR-T cells containing additional co-stimulatory domains (i.e., ICOS, OX40, CD27) remain less well investigated, however, there are apparent trends among co-stimulatory receptor superfamilies which largely influence the metabolic state of the CAR-T cell population. ICOS-mediated activation of the mTOR complex promotes glucose uptake and glycolytic metabolism, analogous to CD28ζ CAR-T cells [191]. The prevalent similarities among metabolic states are likely a result of conserved downstream activation of PI3K/Akt pathway driven by both CD28 and ICOS. TNF-R superfamily molecules, OX40 and 4-1BB, similarly demonstrate an increased CAR-T cell dependency on mitochondrial metabolism. Gene set enrichment analyses have shown that OX40-co-stimulated CAR-T cells are selectively enriched for oxidative phosphorylation pathways, however, functional metabolic readouts to validate unbiased phenotypic profiling remain to be reported (Table 1) [192]. 

Globally, phenotypic evaluations largely correspond with the dominant metabolic processes that are heavily relied upon by specific CAR-T cell designs, which also influence longevity and persistence of the infused population as to be discussed in Section 4.3 [173,174]. 

### 4.2. Response Kinetics

The choice of co-stimulatory domain has been shown to influence the potency of the CAR-T cell response in an in vivo setting [22,193]. When comparing kinetics of the two most well characterized CAR-T cell designs, 4-1BB and CD28, in a model where a sub-optimal dose of CAR-T cells is intentionally infused [44,194], CD28ζ CAR-T cells induced more rapid tumor regression and completely eliminated tumor cells at a dose that only delayed tumor progression in mice treated with BBζ CAR-T cells [195]. Similar trends have been reported in solid tumor settings [65]. When examining in vivo CAR-T cell proliferation, CD28ζ CAR-T cells undergo a robust proliferative burst followed by rapid contraction and clearance of the CAR-T cell population. This is in part attributed to the induction of IL-2 that is largely driven by CD28 to induce clonal T cell expansion. Comparatively, 4-1BBζ CAR-T cell accumulation occurs more gradually, but consistently surpassed the number of CD28-containing CARs indicating that the 4-1BB co-stimulatory domain catalyzes a slower, but more sustained anti-tumor response which is supported by the strong induction of anti-apoptotic molecules. Similar response trends have been observed for additional members of the co-stimulatory family. ICOSζ CAR-T cells targeting mesothelin display tumor elimination kinetics similar to CD28ζ CAR-T cells [173]. However, anti-tumor responses mediated by OX40ζ and CD27ζ CAR-T cells are more gradual in comparison to immunoglobulin superfamily-containing CAR designs, mirroring the responses following 4-1BBζ CAR-T cell infusions [176]. 

The differences in observed response kinetics among second generation CAR-T cell designs are mirrored by notable differences in spatial and temporal regulation of endogenous immunoglobulin and TNF-R superfamily receptors. Molecular level mechanistic studies further demonstrated that CD28-containing CARs signal more rapidly and with increased intensity than 4-1BB containing CARs. Phosphoproteomic analyses reveal that 28ζ and BBζ CAR signaling similarly altered the phosphorylation state of proteins involved in TCR and MAPK signaling, with the primary difference being lower intensity and rate of activation in BBζ CAR-T cells [29,71,76]. Similar studies have compared the activation kinetics of CAR T cells containing either CD28 or ICOS co-stimulatory domains, each of which signal through the recruitment of PI3K and activation of the MAP kinases including JNK. Both ICOSζ and CD28ζ CAR-T cells rapidly induced Akt and Erk signaling as evaluated by protein phosphorylation studies, which further supports the similarities observed in in vivo response studies (Table 1) [69].

The functional and mechanistic studies performed thus far further suggest that engagement of CD28-family molecules support an immediate, potent anti-tumor response whereas TNF-R superfamily molecules, namely, 4-1BB initiate lower-intensity signaling cascades which may contribute to gradual effector responses and sustained CAR-T cell proliferation. The field’s understanding of signaling initiated by CARs remains in its infancy. We do not fully understand the consequences of engineering modular receptors in an attempt to recapitulate the spatially and temporally regulated series of events that culminate in T cell activation. It has become increasingly evident that chimeric signaling entities cross-talk with endogenous co-stimulatory and co-inhibitory receptor-mediated pathways, which further complicates our understanding but will become increasingly important in future improvements to CAR-T cell design. 

### 4.3. Persistence and Durability

It is now widely accepted that functional persistence of the infused CAR-T cell population is critical to durable clinical responses and continuing remission in patients [58]. Persistence is commonly associated with specific phenotypic and metabolic characteristics of CAR-T cells, with long-lived populations displaying properties of stem memory/central memory T cells. In contrast, effector/effector memory populations mediate a potent anti-tumor effect, but subsequently undergo activation induced cell death. Considering each of the clinical trials investigating one of the now several FDA-approved CAR-T cell product, patients who receive 4-1BB-based products generally showed enhanced persistence of the infused product, with circulating CAR-T cells detectable 12–24 months after infusion [49,50,52,196,197]. Despite high frequencies of complete responses in clinical trials, many patients who receive CD28-based CAR-T cell products exhibited short-lived persistence with CAR-T cells falling below detectable levels within 5–6 months [198,199,200]. However, long-term persistence of circulating CD28ζ CAR-T cells was seen in patients who remained in remission after 12–24 months [201,202], which indicates that factors beyond co-stimulation contribute to the striking differences in reported CAR-T cell persistence. Largely, clinical data align with the characterization of 4-1BB and CD28-containing CAR-T cells in regard to their surface phenotypes and dominant metabolic processes [60,171,172].

Preclinical investigations into CAR-T cell persistence largely mirror the clinical data as BBζ CAR-T cells demonstrate improved persistence in comparison to 28ζ CAR-T cells in various murine models [65,70,195,203]. However, additional preclinical investigations into other co-stimulatory receptor domains have been reported. Inclusion of CD27 within the CAR enhanced survival of the adoptively transferred population and improved memory T cell formation in a xenograft model [174]. Furthermore, ICOS-containing CD4^+^ CAR-T cells demonstrated enhanced persistence when compared to both 28ζ and BBζ CAR-T cells, but more importantly, supported the persistence of co-infused CD8^+^ CAR-T cells, providing compelling evidence that an optimal co-stimulatory signal may differ for CD4^+^ and CD8^+^ CAR-T cells [69]. Finally, αBCMA OX40ζ CAR-T cells vastly outperformed both CD28ζ and 4-1BBζ CAR-T cells when evaluating CAR-T cell persistence in peripheral blood in an immunodeficient model of multiple myeloma (Table 1) [192]. Considering the wealth of preclinical data evaluating the influence of co-stimulatory molecules on CAR-T cell persistence, TNF-R superfamily molecules (4-1BB, OX40 and CD27) drive the generation of memory-like cells and initiate signaling cascades of reduced magnitudes, which temper robust responses but promote a sustained population in vivo. Immunoglobulin superfamily member, CD28, drives an effector CAR-T cell population with a rapid kinetic response leading to short-lived durability of the infused population. However, ICOS, which lacks the distal PYAP motif, illustrates a similar kinetic response to CD28ζ CAR-T cells, but results in a population poised to persist longer in vivo. These data support the idea that both the quality and quantity of signaling may be critical in defining an optimal co-stimulatory signal for effective anti-tumor function and robust, durable protection.

There are several caveats to studying persistence in the preclinical setting. Often, persistence is limited to a few days to weeks following infusion and is measured in a range of anatomical locations including peripheral blood, bone marrow and spleen, depending upon the model used. In addition, differences in persistence can be confounded by varying response kinetics, as previously discussed. In the case of BBζ CAR-T cells, initial responses are more gradual relative to CD28, and often illustrate residual tumor burden which can provide lingering stimulation to the CAR-T cell population. To investigate intrinsic differences in persistence resulting from the co-stimulatory domain, timepoints should be determined by the level of disease burden and not by time post-infusion [195,204]. 

### 4.4. Clinical Efficacy and Associated Toxicities

Over the last 10 years a significant amount of data surrounding B cell leukemia and lymphoma patient responses to CD19-targeted CAR-T cell therapy has accumulated, which has established a paradigm for treating additional disease indications, notably multiple myeloma. Sufficient clinical efficacy data exists to compare differences in responses between CD19-targeted 28ζ and BBζ CAR-T cell products. However, differences among CAR-T cell products including the viral vector used for delivery (γ-retroviral vs. lentiviral), hinge and transmembrane domains, and also patient-specific factors such as disease state at the time of treatment, lymphodepleting chemotherapy regimen employed, and the dose of CAR-T cells infused all confound conclusions gleaned from clinical data [9,204]. In the case of B cell lymphoma, which encompasses diffuse large B cell lymphoma, mantle cell lymphoma, follicular lymphoma and high grade B cell lymphoma, complete response (CR) rates following treatment with CD19-targeted CD28ζ CAR-T cells range from 53–67% [201,202,205,206,207] and 40–62% for BBζ CAR-T cell products [49,52,197,208,209]. Similar trends are observed in patients with B cell ALL with CR rates spanning 70–86% and 69–92% for 28ζ and BBζ CAR-T cells, respectively [50,196,200,210,211,212,213,214,215]. Systematic meta-analyses of published clinical trial data have reported that the proportion of patients who achieve a CR when comparing 28ζ and BBζ CAR-T cell products was not significantly different in both B cell leukemia (n = 953) [216] and B cell lymphoma (n = 61) [217] disease settings. 

Response durability, or progression free survival (PFS) is an additional parameter used to evaluate therapeutic efficacy. The reported duration of follow-up is highly trial-dependent and in the context of CD19-targeted CAR-T cells has spanned from 12 months to nearly a decade. In several instances, median PFS is not reached at the time of reporting. Across trials in various B cell lymphoma subsets and chronic lymphocytic leukemia, median PFS for patients treated with both CD28ζ and 4-1BBζ CAR-T cell products spans from 8–36 months [49,52,197,201,205,206,207,209,218,219]. However, there is strong evidence in support of response durability to both 28ζ and BBζ second generation CAR-T cells with a subset of patients who remain in ongoing remission 7–10 years after treatment [58,218]. 

BCMA-targeted CAR-T cells have also been successful in the treatment of relapsed/refractory multiple myeloma, which led to two recent FDA approvals, ciltacabtagene autoleucel and idecabtagene vicleucel. CR rates following treatment with BCMA 4-1BBζ CARs range from 31–78.6%, and from 18.6–63% following treatment with BCMA 28ζ CARs [220,221]. Given the precedence set by CD19-targeted CAR-T cells and the ability of 4-1BB to extend the lifetime of CAR-T cells in vivo, many BCMA targeted CAR-T cell products under investigation contain a 4-1BB co-stimulatory domain [6,7,53,222,223,224]. Specifically, it is critical to note that the number of patients who have received BCMA 28ζ CAR-T cells as part of a clinical trial is nearly one fifth of those who have received 4-1BB co-stimulated CAR-T cell products [9]. Despite similar trends in overall response rates across BCMA CAR-T cell products, patients treated with CAR-T cells containing a CD28 co-stimulatory domain illustrated a shorter median PFS (8 months) than those treated with 4-1BB co-stimulated CAR-T cell products (12.2 months) [9]. The median PFS achieved for the two FDA approved BCMA CAR-T cell products were 12.1 months and 19.9 months, for idecabtagene vicleucel and ciltacabtagene autoleucel, respectively [7,8,53,54,55,222].

Despite stark differences in response kinetics and CAR-T cell persistence, the choice of co-stimulatory domain does not translate into striking difference in clinical response. However, CR and PFS data summarized herein demonstrate that there is room for improvement in CAR design. Therefore, further clinical investigation is needed to directly compare second generation CD28ζ and 4-1BBζ CAR-T cell designs, which may reveal a preferential co-stimulatory signal depending on the tumor type or other extrinsic factors.

Safety remains a clinical concern for all patients undergoing CAR-T cell therapy. Prominent CAR-T therapy toxicities include cytokine release syndrome (CRS) and neurotoxicity, or CAR-T cell related encephalopathy syndrome (CRES), which result from a strong inflammatory response shortly after CAR-T cells are infused. The spectrum of manifestations of CRS range from low grade fever, hypotension and cytopenia to respiratory and cardiovascular failure and in severe cases can be fatal [225]. Low grade CRES typically presents with confusion, disorientation and impaired language and handwriting. However, more severe CRES can cause seizures, motor weakness and increased intracranial pressure [226,227]. The onset of toxicity is mediated by the robust activation of CAR-T cells upon their encounter with antigen expressing tumor cells and the release of cytokines including IFN-γ, IL-2, IL-6, TNF-α, which can further activate bystander immune cells including monocytes and macrophages to secrete additional cytokines and chemokines including IL-10, IL-6, IL-8, CXCL9, CXCL10, MIP-1α and MIP-1β [196,201,205,210,219,228,229,230,231]. In the case of CRES, it is thought that serum cytokines, namely IL-6, diffuse into the brain. However, there is also evidence that CAR-T cells can traffic into the central nervous system to induce a localize state of inflammation [210,219,226,227,232,233]. 

The degree of severity of CRS is positively correlated with the level of tumor burden at the time of treatment, and patient response to therapy [231]. In addition, comparisons among clinical investigations demonstrate a role for the co-stimulatory domain on systemic toxicity profiles. As discussed previously, the kinetics and magnitude of response when comparing CD28ζ and BBζ CAR-T cell products is striking. Unsurprisingly, CD28-containing CARs mediated a robust spike in serum cytokine levels between 3–5 days post infusion, while BBζ CAR-T cells exhibited a significant delay with peak levels occurring 10–21 days after infusion [231,234,235]. The contribution of a robust CD28 signal is thought to explain, in part, differences among incidence and severity of CRS and CRES [204] as well as observations in median time to symptom onset (2 days with 28ζ products vs. 3–5 days with BBζ products) [49,52,201,202]. Across multicenter clinical trials, the vast majority (>85%) of patients treated with CD28ζ CAR-T cell products presented with CRS of any grade as compared to 30–60% of patients treated with BBζ products, and a higher frequency of these patients required intervention with tocilizumab (IL-6 receptor antagonist) [201,202,207,210,211,225,236]. Incidence and severity of neurologic toxicity follow similar patterns as CRS implicating CD28ζ CAR-T in more severe CRES. However, recent findings attributed differences in toxicity to the hinge and transmembrane domains when evaluating CAR-T cell designs in a head-to-head comparison [206,207]. Replacement of the CD28 transmembrane domain with a CD8α domain reduced cytokine levels and improved clinical toxicity profiles with retention of anti-tumor activity. These studies further demonstrate that design of modular components of the CAR can have profound clinical effects beyond the primary anti-cancer objective of CAR-T cell therapy (Table 1).

## 5. Future Perspectives: Harnessing Co-Stimulation to Enhance CAR-T Cell Efficacy 

The influence of co-stimulatory domains on CAR-T cell function has become a major focus of the CAR-T cell field. Specifically, the field lacks an understanding of how immunoglobulin family co-stimulatory molecules, namely CD28, drives CAR-T exhaustion and limited persistence. Whether these functional differences are driven by variations in quantitative signal magnitude or qualitive differences among Ig and TNF-R superfamily signal transduction pathways (Figure 2) remains elusive, with evidence from multiple studies supporting contrasting theories [34,59,72,73,74]. Within the last few years, several high impact studies have illustrated improved persistence and durability of CD28ζ CAR-T cells when co-stimulatory signaling by CD28 was disrupted through targeted mutation of specific amino acid residues within CD28 signaling domains within the CAR. 

Mutation of the asparagine within the membrane proximal CD28 signaling motif (YMNM → YMFM) resulted in a less-differentiated CAR-T cell product with enhanced persistence [73]. Deploying this mutation abrogated binding of Grb2 at the proximal motif thereby reducing PLCγ1 activation and downstream NFAT overactivation (Figure 2), which has been implicated in promoting CAR-T cell exhaustion. Transcriptionally, CD28^ΔYMFM^ζ CAR-T cells resembled Th17 cells displaying gene signatures consistent with reduced T cell differentiation and enhanced stem-like properties, a phenotype similar to ICOS-co-stimulated CAR-T cells as discussed in Section 4.1.1 (Figure 3A) [73,173].

Employing a similar approach, Boucher et al. designed a CAR construct in which only the distal proline motif (PYAP) of the CD28 signaling domain remained functional. Mutation of the proximal tyrosine (YMNM → FMNM) and the middle proline (PRRP → ARRA) signaling domains impaired downstream PI3K and Itk activation (see Section 3.1.1.) resulting in a CAR-T cell product with reduced expression of NFAT exhaustion-related genes and reduced chromatin accessibility at their genomic loci [74]. Similar to the CD28^ΔYMFM^ζ mutants, dual mutation of the membrane proximal and middle CD28 domains improved CAR-T cell persistence and survival in vivo (Figure 3B). Collectively, these studies demonstrate a qualitative role for co-stimulation in altering CAR-T cell function, which may be further complicated by compensatory pathways in CD28-mediated signal transduction [79,94,99,104]. 

Thus far, we have almost exclusively focused on the contributions of the encoded co-stimulatory domain despite the maintained surface expression of endogenous co-stimulatory receptors, including CD28, on human CAR-T cells [220]. In the context of hematologic B cell malignancies (B cell lymphomas, leukemias and multiple myeloma), which express CD28 ligands CD80 and/or CD86, CAR-T cells likely receive additional co-stimulatory signals upon recognition of the target cancer cell and/or upon interaction with antigen presenting cells present in lymph nodes (lymphoma), circulation (leukemia) and bone marrow (myeloma/leukemia). Of particular interest is a recent study from Renier Brentjens’ group showing that overstimulation through the endogenous plus CAR CD28 signaling entities, when combined with a third signal initiated by CAR-T secreted IL-12, drove CAR-T cell dysfunction [72]. In this IL-12 armored CAR-T cell system, loss of endogenous CD28 signaling in IL-12 secreting 28ζ CAR-T cells enhanced their functional capacity and contributed to an in vivo survival advantage (Figure 3C). Furthermore, the authors demonstrate that detrimental levels of co-stimulation are not specific to CD28, as excessive stimulation through 4-1BB receptors also drove CAR-T cell dysfunction, providing evidence in support of an optimal magnitude of co-stimulatory signaling to achieve desirable functional outputs.

There now exists sufficient evidence in the CAR-T cell field that employing an optimally balanced co-stimulatory signal is critical for an effective, durable anti-tumor CAR-T cell response [59,69,72,73,74]. Further investigation is needed to understand the contribution of crosstalk between CAR driven co-stimulatory signals and endogenous co-stimulatory or co-inhibitory receptor signals to second generation CAR-T cell function. Additional confounding signaling crosstalk is introduced when CAR-T cells are armored with stimulatory cytokines (Figure 3C). More complete understanding of the consequences of such crosstalk could allow future CAR designs to preserve beneficial attributes of less differentiated T cells while simultaneously limiting the severity of CAR-T cell-induced toxicities. Clinical testing of CAR-T cells containing mutagenized CD28 domains (Figure 3A,B) and/or incorporating a third cytokine signal (Figure 3C) would be a promising first step towards this ultimate goal.

In summary, co-stimulatory signals are essential for the survival, function and potent anti-tumor activity of CAR-T cells in cancer patients. Despite strong headway made into understanding the implications of co-stimulation in a synthetically engineered T cell, continued research efforts are essential to defining the optimal co-stimulatory signal or signals to employ in CAR design. 

## Figures and Tables

**Figure 1 biomolecules-12-01303-f001:**
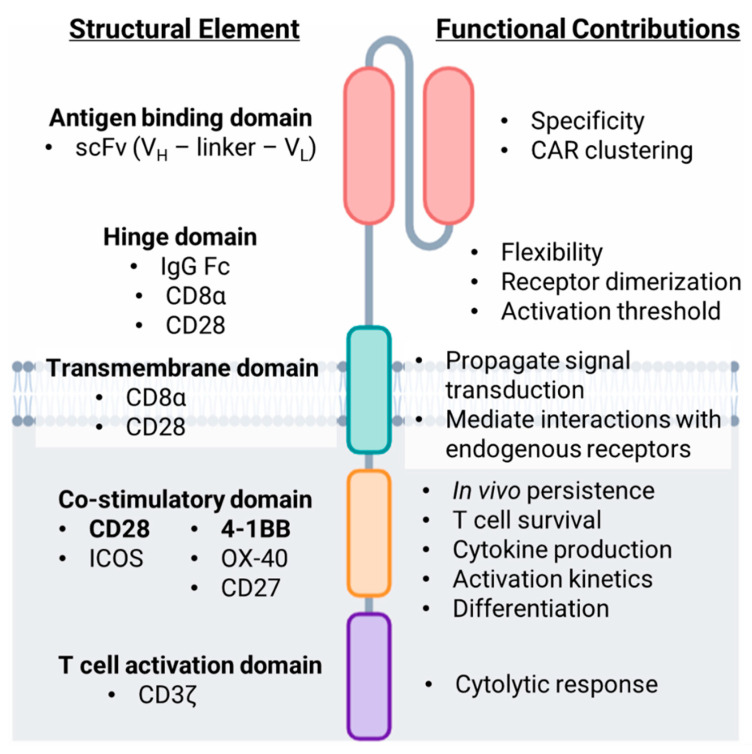
Structure of second-generation Chimeric Antigen Receptor (CAR).

**Figure 2 biomolecules-12-01303-f002:**
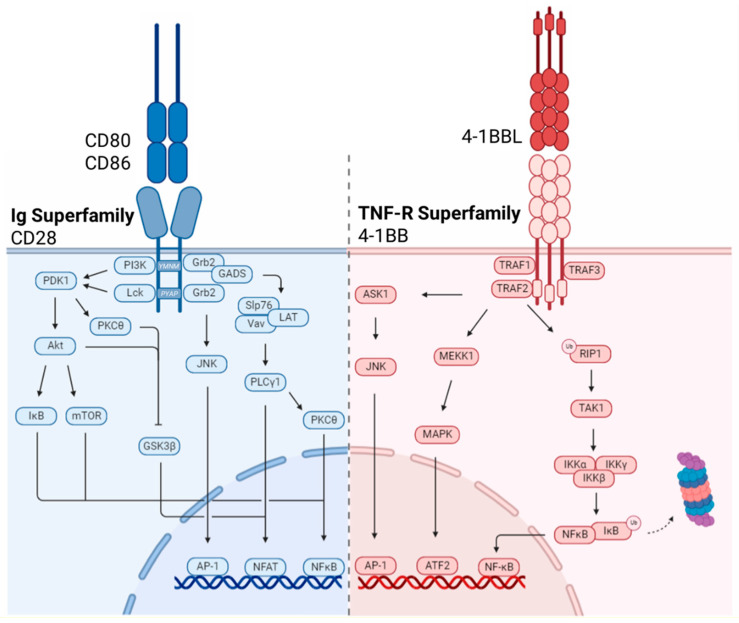
**Co-stimulatory signaling pathways of the Immunoglobulin and TNF-R superfamilies.** CD28 binds to B7 family members CD80 and CD86. Interaction with these ligands initiates the recruitment of adaptor molecules including PI3K, Lck and Grb2. PI3K and Lck activate PDK1 which diverges to activate both canonical NF-κB through Akt activation and NFAT through GSK3β inhibition. The binding of Grb2/GADS initiates the SLP76/LAT/Vav complex formation which activates PLCγ1 leading to NFAT-dependent gene transcription. PLCγ1 also activates PKCθ, which regulates activation of NF-κB. Grb2 can also localize independently at the distal motif to mediate MAPK and JNK activation, promoting nuclear translocation of the AP-1 complex. 4-1BB binds to its sole ligand 4-1BBL, which induces the recruitment of TRAF molecules. TRAF2 mediates RIP1 ubiquitination leading to the recruitment of TAK1 and activation of the IKK complex. IKK activation promotes the ubiquitin-mediated proteasomal degradation of IκB, releasing NF-κB subunits from an inhibitory state. Independent of NF-κB signaling, TRAF2 also activates upstream MAPK, MEKK1 and ASK-1, which initiate the p38 MAPK signaling cascade and the JNK/SAPK pathway, respectively. Transcriptional activity induced by both Ig and TNF-R signaling induce anti-apoptotic signals, T cell proliferation and cytokine production (IL-2).

**Figure 3 biomolecules-12-01303-f003:**
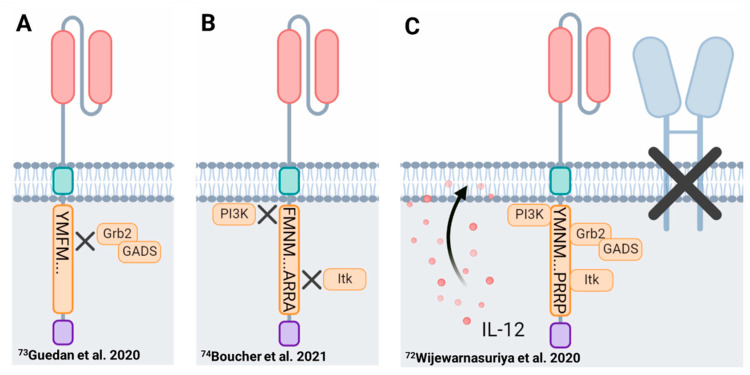
Targeting CD28 to optimize co-stimulatory signals. (**A**), (**B**). Site-directed mutagenesis of amino acid residues in the CD28 domain abrogate binding of adaptor signaling molecules. (**C**). Genetic deletion of the endogenous CD28 receptor eliminates additional co-stimulatory signals delivered to CAR-T cells in vivo. Reductions in CD28-mediated activation signals preserve stem memory/central memory phenotypes, reduce expression and chromatin accessibility of exhaustion-associated genes and promote enhanced CAR-T cell persistence in preclinical models [72,73,74].

**Table 1 biomolecules-12-01303-t001:** Comparison of the functional attributes of co-stimulatory domains in second generation CAR-T cells.

Co-Stimulatory Domain	Receptor Family	Differentiation	Exhaustion	Metabolic Landscape	Kinetics	Persistence	Toxicity
* **CD28** *	Ig Superfamily	Effector memory	Prone to exhaustion	Aerobic glycolysis	Rapid signaling kinetics, greater phosphorylation intensity, greater cytokine release, rapid tumor regression	Short-lived	Rapid symptom onset (within 48 h. of infusion), greater frequency and severity of CRS, greater frequency of patients require intervention
* **ICOS** *	Ig Superfamily	T_H_17 polarization (self renewal and stem-like properties)	Less susceptible to exhaustion	Aerobic glycolysis	Rapid signaling kinetics, greater phosphorylation intensity, greater cytokine release, rapid tumor regression	Long-lived	Pending clinical investigation
* **4-1BB** *	TNF-R Superfamily	Central memory	Less susceptible to exhaustion	Oxidative phosphorylation and fatty acid oxidation	Slower, less intense signaling, reduced cytokine release, gradual tumor regression	Long-lived	Delayed symptom onset (3–5 days of infusion), lower frequency and severity of CRS, lower frequency of patients require intervention
* **OX40** *	TNF-R Superfamily	Central memory	Less susceptible to exhaustion	Oxidative phosphorylation (transcriptomic level analysis)	Reduced cytokine release, gradual tumor regression	Long-lived	Pending clinical investigation
* **CD27** *	TNF-R Superfamily	Central memory	Less susceptible to exhaustion	---	Rapid tumor regression	Long-lived	Pending clinical investigation

## Data Availability

Not applicable.

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
