# Peer review of "Co-Stimulatory Receptor Signaling in CAR-T Cells"

_biomolecules, 2022, doi:10.3390/biom12091303_

Round 1
Reviewer 1 Report
Strengths:
1. well-written article covering an important topic – importance of co-stimulatory signaling and impact of these co-stim molecules on clinical outcomes and relating this to functional consequences of incorporation of different co-stim signals.
2. introduces multiple co-stim signal molecules and does a thorough job of mapping out associated signaling cascades while discussing the functional consequences of each signal (essentially how they differentially contribute to proliferation, cytokine production, promotion of exhaustion, etc).
3. The authors nicely compare and contrast the clinical and correlative data available for CAR T cell constructs incorporating CD28 versus 4-1BB costimulatory components. Discussion topics include effects on T cell phenotype, differentiation, cellular metabolism, response kinetics, and persistence. They also compare impact of CD28 vs 4-1BB costimulation in terms of clinical efficacy and toxicity profiles.
Weaknesses:
1. The authors' perspective would seem to be derived more from CD19-directed CAR T cells, less so on BCMA-directed CAR products. This is not necessarily a problem but if this is the authors' intent, they should acknowledge this focus on CD19-targeting products.
• The statement they make regarding BCMA CAR T cell therapies in myeloma, that PFS is much reduced at 3-11 months, is founded on old data and the authors fail to cite the current literature including publications of clinical data for cilta-cel, LCAR-B38M and CT103A. All of these products have generated data supporting PFS >12 months.
• The statement “ScFv molecules are commonly derived from mouse monoclonal antibodies” is largely true but not entirely. CT103A and CT053 are fully human scFv constructs. LCAR-B38M is derived from alpaca. Again, this contributes to the sense that the authors are drawing their information largely from the CD19 CAR T cell constructs in writing this manuscript but are not quite as well-versed in the BCMA CAR T cell products developed for myeloma.
2. While the authors invest considerable effort to describe signaling cascades and functional consequences of stimulation of ICOS, CD27 OX40, they never seem to circle back to discuss the relevance of these and their potential to improve CAR T cell function.
Suggestions:
1. include a table to identify the CAR T cell constructs including relevant details from which this review is drawing its conclusions and the facts supporting these conclusions.
2. If not planning to review preclinical data with ICOS, CD27 and/or OX40, or to at least discuss potential impact for development in the future directions section, for instance, I would probably remove the sections detailing their signaling. Afterall, this is a manuscript reviewing CAR T cell co-receptor signaling, not general T cell signaling.
Reviewer 2 Report
It is a very well written review article summarizing the different co-stimulatory domains used in the CAR T cell field. The authors also comprehensively discuss the different domains and their functional implications with respect to signaling, CAR T cell phenotype and clinical outcomes. There are a few minor comments which could be clarified:
1. It’s well known that 2nd Gen CARs with one costimulatory domains are only used now. However, 3rd gen CARS with 2 co-stimulatory domains have existed and are worth mentioning and discussing why they are not used anymore and why 2nd gen CARS are preferred.
2. The authors bring forth the advantages and the differences between the different co-stimulatory domains (mainly CD28 and 41BB). However, the disadvantages of using one or the other domain also should be pointed out clearly. For e.g. it is known that 41BB domain prevents tonic signaling and T cell exhaustion when compared to CD28 domain when CARs cluster (PMID: 25939063). Such disadvantages can be pointed out.
3. The differences discussed in the text (Section 3 and 4) can be summarized as a table. Will be helpful for the reader to understand the better.
4. Lines 157 to 162. Is it know how mutating the ITAMS preserves memory phenotype? If so elaborate.
5. The cytoplasmic signaling domains in Fig 2 can be illustrated better. It will be easier to follow the text if the different domains membrane proximal tyrosine-rich motif (YMNM), the proximal proline-rich motif (PRRP) and the distal proline-rich motif (PYAP) and their respective binding partners are clearly pointed out as described in the text.
